# Quiescence-Origin Senescence: A New Paradigm in Cellular Aging

**DOI:** 10.3390/biomedicines12081837

**Published:** 2024-08-13

**Authors:** Guang Yao

**Affiliations:** 1Department of Molecular & Cellular Biology, University of Arizona, Tucson, AZ 85721, USA; guangyao@arizona.edu; 2Arizona Cancer Center, University of Arizona, Tucson, AZ 85719, USA

**Keywords:** quiescence, senescence, quiescence deepening, dormancy state continuum, Rb–E2F switch threshold, geroconversion

## Abstract

Cellular senescence, traditionally viewed as a consequence of proliferating and growing cells overwhelmed by extensive stresses and damage, has long been recognized as a critical cellular aging mechanism. Recent research, however, has revealed a novel pathway termed “quiescence-origin senescence”, where cells directly transition into senescence from the quiescent state, bypassing cell proliferation and growth. This opinion paper presents a framework conceptualizing a continuum between quiescence and senescence with quiescence deepening as a precursor to senescence entry. We explore the triggers and controllers of this process and discuss its biological implications. Given that the majority of cells in the human body are dormant rather than proliferative, understanding quiescence-origin senescence has significant implications for tissue homeostasis, aging, cancer, and various disease processes. The new paradigm in exploring this previously overlooked senescent cell population may reshape our intervention strategies for age-related diseases and tissue regeneration.

## 1. Introduction

Cellular senescence, a state of permanent cell cycle arrest, has been a focal point of biomedical research for decades [1]. It is typically triggered when growing cells are overwhelmed by extensive stresses and damage, such as telomere erosion, oncogene activation, oxidative stress, and DNA damage, to a degree beyond their repair capacity [2,3]. Cells enter senescence and permanently halt proliferation to avoid deleterious consequences like malignant tumorigenesis and fibrotic response [4,5]. The hallmarks of senescent cells include permanent cell cycle arrest, senescence-associated secretory phenotype (SASP), macromolecular damage, deregulated metabolism [6], and often enlarged cell size and other morphological changes [2,7]. This conventional view has shaped our understanding of senescence primarily as an extreme stress response in proliferating and growing cells.

In contrast, cellular quiescence is a reversible state of cell cycle arrest. Under physiological growth-limiting conditions, such as mitogen withdrawal or contact inhibition, cells exit the cell cycle and enter quiescence [8]. Quiescent cells are typically associated with low metabolic activity and reduced cell size and do not exhibit obvious deleterious phenotypes associated with senescence, such as SASP and severe damage [8,9]. Upon restoration of growth-permissive conditions (e.g., reintroducing growth factors or alleviating contact inhibition), quiescent cells can resume proliferation, which is a key distinction from senescent cells [8,9]. Quiescence is considered a physiological state that allows cells to maintain cellular viability and integrity under growth-limiting conditions, playing a crucial role in tissue homeostasis and stem cell maintenance [9,10]. This mechanism is proposed to safeguard cells from senescence [11,12]. 

Recent research in the past decade, however, has begun to challenge this traditional paradigm by demonstrating that senescence can originate directly from quiescent cells in various contexts [13,14,15,16]. This phenomenon, which we term “quiescence-origin senescence”, suggests a fluid continuum between quiescence and senescence with profound implications for tissue homeostasis, aging, and disease processes. Notably, proliferating cells account for less than 1% of the thirty-seven trillion cells in the human body at any given time [17], suggesting that quiescence-origin senescence may underlie a large proportion of previously overlooked or misclassified senescent cells. 

This brief review synthesizes recent findings to provide a new framework for understanding senescence from the perspective of cell dormancy–state plasticity, aiming to stimulate further related research into novel intervention strategies for age-related diseases, cancer, and regenerative medicine.

## 2. The Quiescence–Senescence Continuum: A Framework for Understanding Quiescence-Origin Senescence

### 2.1. Direct Transition from Quiescence to Senescence

Mounting evidence in the past decade demonstrates a direct transition to senescence in quiescent cells under stress or with aging. One of the first reports of this phenomenon was in geriatric mice, where quiescent satellite cells (muscle stem cells, MuSCs) lose regenerative capacity and become irreversibly arrested [13]. The same observation was confirmed in geriatric human muscle, suggesting a conserved mechanism contributing to age-related decline in muscle regenerative capacity [13]. The quiescence-to-senescence transition in geriatric MuSCs was linked to the epigenetic derepression of p16^INK4a^, which is a key regulator of senescence [13].

Further research revealed that p16^INK4a^ derepression in geriatric MuSCs was triggered by accumulated reactive oxygen species (ROS) resulting from failed autophagy [18]. Restoring autophagy prevents the transition to senescence in quiescent MuSCs [18]. Similarly, in quiescent human hematopoietic stem progenitor cells (HSPCs), stress-induced ROS accumulation can elevate the p16^INK4a^ level and drive the transition to senescence [14]. 

In vitro studies provide further evidence for the direct quiescence-to-senescence transition. Long-term quiescent human fibroblasts in culture (100–150 days) accumulate DNA damage and transit into senescence [15]. Our recent work showed that quiescent rat embryonic fibroblasts (REFs) with impaired lysosomal autophagy can directly enter a senescence-like state characterized by irreversible cell cycle arrest, β-galactosidase activity, and cellular hypertrophy [16]. These in vitro results strengthen the findings of in vivo studies by showing that cells maintained in quiescent conditions throughout the experiments eventually transitioned to senescence, bypassing cell proliferation (which can be difficult to track in vivo, e.g., throughout the entire geriatric phase). Together, these in vivo and in vitro studies provide compelling evidence for the direct quiescence-to-senescence transition, resulting in quiescence-origin senescence.

### 2.2. Quiescence Deepening into Senescence: A Continuum Model

The observation that cells can transition from quiescence to senescence aligns with the concept that quiescence is not a uniform state but rather a spectrum of dormant depths. Specifically, deeper quiescent cells are more resistant to cell cycle re-entry, and they resume proliferation less frequently or with a longer delay than shallower quiescent cells given the same growth stimulation [8,19,20,21,22,23,24,25,26,27,28,29,30,31,32,33,34,35]. Cells in deep quiescence generally exhibit decreased transcription, reduced RNA and protein levels, and increased chromatin condensation compared to cells in shallow quiescence [22,36,37,38]. Recent studies have further revealed the changes in the cellular profiles of gene expression, epigenetic modification, and metabolism that occur during the process of quiescence deepening [8,16,39,40,41]. It is important to note that deep quiescent cells, by definition, still retain the potential for cell cycle re-entry and can resume proliferation given a sufficiently strong and sustained growth stimulus, which is distinct from senescent cells. 

Quiescence deepening has been observed under extended quiescence in culture or along the aging process in vivo across mammalian cell types and tissues [8,19,20,21,22,23,24,25,26,27,28,29,30,31,32,33] as well as in yeast [34] and C. elegans [35]. Consistent with quiescence depth being a continuum, subpopulations of quiescent muscle, neural, and hematopoietic stem cells, upon tissue injury or in response to systemic factors, can move to shallower (primed or G_Alert_) quiescence, bearing a closer resemblance to active cells [42,43,44,45]. Further extending the cell dormancy state continuum, senescent cells also exhibit varying “depths” with different cellular characteristics [46,47,48].

Our recent work directly links quiescence deepening to senescence entry. We found that while enhancing lysosomal autophagy activity pushes REFs toward shallow quiescence, decreasing lysosomal autophagy drives REFs into progressively deeper quiescence [16]. When lysosomal autophagy is severely hampered (with chloroquine treatment, for example), deep quiescent (yet still reversible) cells transition into irreversible senescence [16]. This suggests that lysosomal autophagy activity acts as a “dimmer switch” that continuously adjusts cellular dormancy depth from shallow quiescence to deep quiescence and finally to senescence [16,41]. Furthermore, the transcriptomic profile changes during quiescence deepening in REFs resemble those of senescence entry with similar expression changes in “senescence core signature” genes [16,49]. A gene signature derived from quiescence deepening in REFs, the quiescence depth score (QDS), can correctly predict a wide array of senescent and aging cell types [16,50]. Together, these findings suggest that quiescence deepening represents a continuum transition trajectory from proliferation to senescence, which is a model increasingly recognized in the field [9,51,52,53,54,55].

Supporting the quiescence–senescence continuum concept, the levels of multiple widely used senescence biomarkers (e.g., SA-β-gal, LAMP1, IL8, 53BP1, p21, Lamin B1, and cell size) were found to increase gradually with the duration of cell-cycle withdrawal and cannot distinguish senescence from deep quiescence [56]. This quiescence–senescence continuum could help explain some normal physiological functions of cellular senescence [57], as some of the senescent cells identified based on biomarker intensities may instead be deep quiescent cells.

## 3. Mechanisms Underlying Quiescence Deepening and Quiescence-Origin Senescence

The cellular processes underlying quiescence deepening and transition to senescence likely involve a complex interplay between various cellular mechanisms and signaling pathways. Yet, a converging theme has started to emerge from recent studies. 

### 3.1. Trigger: The Lysosome–Autophagy Axis

The lysosome–autophagy axis has emerged as a critical driver of quiescence deepening and transition to senescence across various cell types. Autophagy involves autophagosomes engulfing cellular components and fusing with lysosomes, where lysosomal enzymes degrade the contents and recycle the macromolecules. This lysosome–autophagy activity acts as a “dimmer switch” regulating quiescence depth in REFs with its continuously decreased activity leading to progressively deeper quiescence toward senescence [16]. It also underlies the quiescence-to-senescence transition in geriatric MuSCs [18]. 

In both REFs and geriatric MuSCs, ROS accumulation serves as a key effector of the impaired lysosomal autophagy function [16,18]. In REFs, ROS accumulation, without evoking p16^INK4a^, leads to quiescence deepening [16]; in MuSCs, ROS accumulation epigenetically derepresses the p16^INK4a^ locus and leads to the quiescence-to-senescence transition [18]. Similarly, ROS accumulation drives the quiescence-to-senescence transition in HSPCs by elevating the p16^INK4a^ level [14]. 

In neural stem cells (NSCs), the lysosome–autophagy axis, with increased protein aggregates as the main effector, plays a crucial role in quiescence deepening. Aging quiescent NSCs display lysosomal defects, increased protein aggregates, and diminished activation potential [58]. 

Enhancing lysosomal autophagy in various cell types, including REFs, MuSCs, and NSCs, promotes shallow quiescence away from the transition to senescence [16,18,58]. Together, the findings above highlight the lysosome–autophagy axis as a central trigger of quiescence deepening and the quiescence-to-senescence transition. 

### 3.2. Controller: The Rb–E2F Switch Threshold

Our earlier studies demonstrated that the Rb–E2F pathway, a pivotal regulator of the cell cycle in animals and plants [59,60,61,62,63], functions as a bistable toggle switch [64,65]. This Rb–E2F switch converts graded and transient mitogen signals into a binary (OFF/ON) activity of E2F (representing the combined E2F transcriptional activators, E2F1-3a), which underlies the all-or-none quiescence-to-proliferation transition [27,64,65]. The activation threshold of the Rb–E2F switch, “Rb–E2F switch threshold” for short, controls quiescence depth: increasing or decreasing this threshold drives cells to deeper and shallower quiescence, respectively [27,28,66]. The Rb–E2F switch threshold is defined as the minimum growth stimulus required to activate the cellular E2F level to exceed that of non-phosphorylated Rb, which binds to and inhibits E2F in quiescent cells [27,28,66]. We have shown that the Rb–E2F switch threshold can be modulated by multiple factors in the Rb–E2F pathway with varying degrees of sensitivity with those associated with G1 cyclins and cyclin-dependent kinases (CDK) being the most prominent [28]. 

The Rb–E2F switch is the first bistable switch that cells encounter during the transition from quiescence to proliferation in response to mitogens, and it connects to downstream bistable switches and checkpoints regulating S phase entry [67,68]. Given that the Rb–E2F pathway interacts with other quiescence regulatory pathways (e.g., Notch–Hes1, p53–p21, PI3K–Akt, DNA damage response, LKB1–AMPK, and microRNA) [10,27,63], we propose that the Rb–E2F switch serves as a core hub that integrates cellular signals to quantitatively determine the growth stimulation level required for the all-or-none cell cycle re-entry. In parallel, other cellular pathways impinge on the Rb–E2F switch and modulate its switch threshold. This modulatory effect may occur through crosstalk with various gene nodes in the Rb–E2F switch network, most effectively by affecting G1 cyclin and CDK activities in the cell [13,28,69]. Higher thresholds lead to deeper quiescence and, eventually, senescence when physiological growth signals can no longer overcome the threshold [27,41]. Therefore, the difference between deep quiescence and senescence lies primarily in the magnitude of the Rb–E2F switch threshold rather than in fundamentally distinct cellular characteristics, aligning with the quiescence–senescence continuum model.

The lysosomal–autophagy axis discussed above driving quiescence-to-senescence transition in MuSCs directly increases the Rb–E2F switch threshold (by activating p16 that inhibits CDK activity) [13,18]. Similarly, increasing the Rb–E2F switch threshold has been reported to drive deep quiescence in MuSCs via fasting-induced ketone body signaling (by increasing p21 that inhibits CDK) [69]. This threshold mechanism predicts a potential way to reverse the “irreversible” senescence by reducing the Rb–E2F switch threshold. Indeed, studies have shown that senescent cells can be reactivated as such, e.g., by inactivating Rb, p53–p21, or p16 activities [70,71,72,73,74,75] or by the forced G1 cyclin upregulation [76].

### 3.3. Remaining Unknowns and the Heterogeneity in Quiescence-Origin Senescence

Several key areas remain to be explored. The role of mTOR signaling, for instance, presents an intriguing paradox. mTOR is a major driver of proliferation-origin senescence, underlying the process of geroconversion—a process in which mTOR-driven cell growth continues aberrantly after the cell cycle is blocked (e.g., by CDK inhibitors p21^CIP1/Waf1^ and p16^INK4a^) [11,77]. In quiescence, mTOR is inactive and cells lack sustained growth, which has been proposed as a mechanism to protect quiescent cells against senescence [11,77]. This questions the involvement of mTOR in quiescence deepening and the quiescence-origin senescence, which aligns with the fact that mTOR activity is linked to shallow quiescence (G_Alert_) [42], not quiescence deepening. Therefore, the quiescence-origin senescence appears different from proliferation-origin senescence from the aspect of mTOR and geroconversion (Figure 1).

Epigenetic regulation plays a crucial role in proliferation-origin senescence, including the formation of senescence-associated heterochromatin foci (SAHF) that repress E2F target genes [78] and changes in DNA methylation and histone modification patterns [79,80]. Epigenetic mechanisms are also involved in quiescence deepening and transition to senescence, as seen in the ROS-mediated derepression of the p16 locus [18] and the role of ketone body-mediated acetylation and activation of the p53 locus [69] in MuSCs as well as the long-observed increase in chromatin condensation in deep quiescent cells [37]. However, a comprehensive understanding of the epigenetic landscape changes during quiescence deepening and the quiescence-to-senescence transition is still lacking, highlighting an important area for future research [40].

The heterogeneity and context dependency of the quiescence-to-senescence transition are also apparent, underscoring the complexity of these cellular processes. For instance, ROS-mediated p16^INK4a^ activation occurs through epigenetic derepression in MuSCs [18] but via p38 MAPK hyperactivation in HSPCs [14]. Blocking p16^INK4a^ activation prevents the quiescence-to-senescence transition only in a subset of cells [13], suggesting the involvement of other molecular effectors driving the quiescence-origin senescence. Furthermore, is lysosomal autophagy always the primary trigger for quiescence deepening and transition to senescence, and does this process always converge onto the Rb–E2F switch threshold—or possibly regulate downstream switches controlling S phase entry [67,68] or downstream E2F target genes necessary for cell proliferation [41,78]? Growing cells exiting from either the G1 or G2 phase of the cell cycle converge onto the same senescent state with a G1-like molecular signature [76]. How about quiescence-origin senescence? What are its molecular markers, and to what extent do they overlap with or differ from the markers of proliferation-origin senescence, beyond the sustained (if not totally irreversible) arrested phenotype (Figure 1)? Future investigations are needed to find the answers.

## 4. Conclusions

Unlike conventional proliferation-origin senescence, quiescence-origin senescence involves the direct transition of quiescent cells to senescence. It represents a previously understudied new type of cellular aging process (Figure 1). Given that the vast majority of cells in the human body are in quiescence, not in proliferation, quiescence-origin senescence is expected to be responsible for a larger proportion of irreversibly arrested senescent cells in our tissues than previously realized.

Understanding the mechanisms underlying the continuum of quiescence-to-senescence transition through quiescence deepening has important implications for multiple areas of biomedical research and clinical practice, such as cancer biology, aging, and regenerative medicine. Future research will reveal both common and cell type-specific mechanisms of quiescence-depth regulation and transition to senescence, identify specific molecular markers, and suggest the optimal strategies to target this group of previously overlooked senescent cells to reshape our approaches to promoting health and longevity.

## Figures and Tables

**Figure 1 biomedicines-12-01837-f001:**
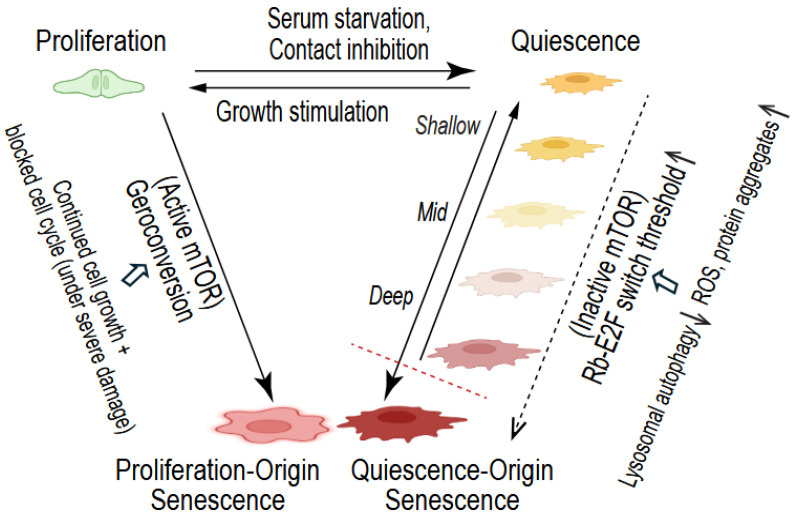
Quiescence-origin senescence. Conventional senescence originates from proliferating and growing cells via mTOR-mediated geroconversion, when cell growth continues aberrantly after the cell cycle is blocked in response to severe stress and damage. In comparison, quiescence-origin senescence originates from quiescent cells via quiescence deepening, accompanied by a progressively increasing Rb–E2F switch threshold in response to decreased lysosomal autophagy and increased ROS accumulation and protein aggregation. Senescent cells are irreversibly arrested under physiological conditions. Deep quiescent cells are reversible but require stronger growth stimulation and take a longer time to re-enter the cell cycle than shallow quiescent cells. The boundary between very deep quiescence and quiescence-origin senescence (the red dashed line) is blurred.

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
