# Peer review of "Quiescence-Origin Senescence: A New Paradigm in Cellular Aging"

_biomedicines, 2024, doi:10.3390/biomedicines12081837_

Round 1

Reviewer 1 Report

Comments and Suggestions for Authors

In the manuscript, the author reported a short summary on quiescence-origin senescence. The induction, regulation and the biological implications of this process is discussed with some examples. The understanding on quiescence-origin senescence may have implications for tissue homeostasis, aging, cancer, and some disease processes. Therefore, the manuscript is interesting. In general, the well-written and could be recommended for publication.

Minor issues

-  - It seems that most references cited are not up-to-date. More recently published studied could be included.

-         - Reference numbers are doubled.

Reviewer 2 Report

Comments and Suggestions for Authors

The review introduces a novel perspective on cellular senescence by focusing on "quiescence-origin senescence". It proposes a continuum model between quiescence and senescence, with quiescence deepening as a precursor to senescence. This challenges the traditional view that senescence arises primarily from cellular stress during proliferation. This review advances our current understanding of senescence mechanisms, which have important implications for tissue homeostasis, aging, cancer, and disease processes.

1. Lines 23&35: The paragraph describes senescence well but briefly mentions quiescence without a detailed explanation. It would be beneficial to clarify how quiescence is distinct from senescence, especially in the context of the "reversible withdrawal" aspect.

2. Line 48: This subheading appears confusing. Please consider revising it for better clarification.

3. Lines 75-105: What specific characteristics differentiate shallow from deep quiescence? Please consider expanding the discussion to help better understand the continuum model.

4. Line 111: Some subheadings are capitalized while others are not. Please ensure a consistent style throughout by standardizing the capitalization of all headings.

5. Lines 153-154: It is stated that other pathways modulate the Rb-E2F switch threshold, but specific examples or mechanisms are not provided. What are the mechanisms through which these pathways modulate the threshold?

6. Figure 1 appears to be oversimplified. Please consider adding more details to the figure to provide a more comprehensive summary of the key concepts.
